# Blackcurrant Anthocyanins Improve Blood Lipids and Biomarkers of Inflammation and Oxidative Stress in Healthy Women in Menopause Transition without Changing Body Composition

**DOI:** 10.3390/biomedicines11102834

**Published:** 2023-10-19

**Authors:** Briana M. Nosal, Junichi R. Sakaki, Manije Darooghegi Mofrad, Zachary Macdonald, Kyle J. Mahoney, Staci N. Thornton, Dave Patel, Joseph Drossman, Elaine Choung-Hee Lee, Ock K. Chun

**Affiliations:** 1Department of Nutritional Sciences, University of Connecticut, Storrs, CT 06269, USA; briana.nosal@uconn.edu (B.M.N.); junichi.sakaki@uconn.edu (J.R.S.); manije.darooghegi_mofrad@uconn.edu (M.D.M.); dave.patel@uconn.edu (D.P.); joseph.drossman@uconn.edu (J.D.); 2Department of Kinesiology, University of Connecticut, Storrs, CT 06269, USA; zachary.macdonald@uconn.edu (Z.M.); kyle.j.mahoney@uconn.edu (K.J.M.); staci.thornton@uconn.edu (S.N.T.); elaine.c.lee@uconn.edu (E.C.-H.L.)

**Keywords:** blackcurrant, anthocyanins, cardiovascular disease, menopause, dyslipidemia, oxidative stress, inflammation

## Abstract

Recent cell and animal studies suggest the potential of blackcurrants (BCs; *Ribes nigrum*) as a dietary agent that may reduce the risk of cardiovascular disease (CVD) by improving dyslipidemia, oxidative stress, and inflammation. This study aimed to examine the effects of BC anthocyanin (ACN) extract supplementation on biomarkers of CVD risk in healthy adult women in menopause transition. The effects of BC ACN supplementation on body composition, fasting blood lipids and biomarkers of inflammation and oxidative stress were evaluated using anthropometric measures and blood samples collected from a pilot randomized controlled clinical trial in peri- and early postmenopausal women. Thirty-eight eligible peri- and early postmenopausal women aged 45–60 completed the entire trial, in which they were randomly assigned into one of three treatment groups: placebo (control group), 392 mg/day (low BC group), or 784 mg/day (high BC group) for six months. The significance of differences in outcomes was tested using repeated-measures ANOVA. Overall, following six-month BC consumption, significantly decreased triglyceride (TG) levels were observed between treatment groups (*p* < 0.05) in a dose-dependent manner. Plasma interleukin-1β (IL-1β) was significantly reduced in a dose and time dependent manner (*p* < 0.05). Significant decreases in thiobarbituric acid reactive substances (TBARS) levels were also observed between treatment groups (*p* < 0.05) in a dose-dependent manner. Six-month change in oxidized LDL was inversely correlated with changes in catalase (CAT) and total antioxidant capacity (TAC) (*p* < 0.05), while C-reactive protein (hs-CRP) change was positively correlated with changes in TG and IL-1β (*p* < 0.01). Together, these findings suggest that daily BC consumption for six months effectively improved dyslipidemia, inflammation, and lipid peroxidation, thus potentially mitigating the risk of postmenopausal CVD development in study participants. Future studies with larger sample sizes and at-risk populations are warranted to confirm these findings.

## 1. Introduction

Cardiovascular disease (CVD) is responsible for causing approximately 35% of deaths in women each year globally; however, it remains understudied, under-diagnosed, and under-treated in women [1], who have a noteworthy increase in the risk for this disease after menopause and typically develop coronary heart disease (CHD) several years later than men. Risk for CVD increases in women at midlife [2], usually coincident with menopause transition. Longitudinal studies of women have significantly contributed to our understanding of the relationship between the menopause transition and CVD risk [3]. By following women during this period, studies have documented patterns of a decrease in estrogen, a natural antioxidant, concurrently with unfavorable alterations in body fat distribution [4], lipids and measures of vascular health, which can increase the risk of postmenopausal CVD development [5]. These reported findings underline menopause transition as a time of accelerating CVD risk and thus emphasizing it as a critical time for implementing intervention strategies.

The search for safe and effective dietary agents that possess estrogen-like effects and provide anti-atherosclerotic, antioxidant, and anti-inflammatory actions has drawn more attention for preventing CVD progression among those in menopause transition due to the high costs and detrimental side effects associated with CVD attenuating drugs [6]. However, women are still underrepresented in clinical trials for CVD risk [7], and intervention studies investigating dietary agents for CVD risk reduction in women during menopause transition are lacking.

Anthocyanin (ACN) intake has been shown to exert anti-inflammatory and antioxidant properties, indicating its potential to attenuate the development of chronic diseases, including CVD [8]. Blackcurrant (BC; *Ribes Nigrum*), specifically, has drawn our attention because we previously found that it contains the largest amount of ACNs among commonly consumed berries [9]. Previous findings suggest that BC possesses a number of health benefits, including alleviating oxidative stress-related conditions [10,11] including dyslipidemia [12] and exerting antioxidant [13,14] and anti-inflammatory effects [15,16,17], which all could contribute to decreased CVD risk. 

To our knowledge, no prior studies have investigated the cardioprotective effects of BC in this high-risk population. Thus, this study aimed to examine whether and how BC supplementation reduces CVD risk. We hypothesized that daily BC ACN supplementation for 6 months will lower the risk of CVD in healthy women in menopause transition by improving blood lipids and biomarkers of inflammation and oxidative stress.

## 2. Materials and Methods

### 2.1. Study Design

#### 2.1.1. Study Overview, Recruitment and Eligibility

The Blackcurrant and Bone Study (Clinical Trials NCT04431960) is a pilot randomized, double-blind, placebo-controlled, 3-arm clinical trial to study the effects of a six-month BC supplementation on bone health and the gut microbiome of peri- and early postmenopausal women (detailed description of study design and bone data are published elsewhere) [18]. Briefly, potential participants were recruited from northeastern Connecticut with advertisements in newspapers, flyers, and emails. Eligible participants were women 45–60 years old, not on HRT for at least one year before beginning the study, and maintaining normal exercise levels (<7 hours/week). Exclusion criteria included those with major chronic disease(s), heavy smokers, any chance or plan of pregnancy, taking prescription medications that may alter bone and calcium metabolism, taking anabolic agents, or heavy alcohol consumption (exceeding 2 drinks/day or a total of 12/week).

#### 2.1.2. Blackcurrant Intervention

All study procedures were performed at the Korey Stringer Institute Human Performance Laboratory at the University of Connecticut Storrs Campus from July 2021 to October 2022, and informed consent was obtained during the initial in-person visit. All eligible participants, upon signing the consent form, completed an interview to collect medical history and dietary behaviors and an initial physical examination to record anthropometric measurements.

After completion of the initial visit, all subjects underwent a 2-week equilibration period and were provided a calcium citrate caplet to ingest daily for the entire duration of the study, including 400 mg calcium and 500 IU vitamin D (Bayer AG, Leverkusen, Germany), to avoid potential bone deterioration caused by deficiency of these two essential bone nutrients. After equilibration, participants were assigned randomly into one of three groups: (1) control (one placebo capsule/day); (2) low BC (one 392 mg BC capsule/day); or (3) high BC (two 392 mg BC capsules/day, 784 mg total) for 6 months. Supplement distribution was performed blindly during study visits 1 (baseline) and 2 (3 months). The BC powder was provided by Just The Berries, New Zealand (Just the Berries PD Corporation, Los Angeles, CA, USA). The placebo and BC extract were encapsulated and packaged in coded containers based on dose (Beehive Botanicals, Hayward, WI, USA). Placebo capsules were identical looking but devoid of ACN.

Participants completed an additional consent form to confirm eligibility and were required to have a negative urine pregnancy test before the dual x-ray absorptiometry (DXA) scan during study visits 1 (baseline) and 3 (month 6). A physical examination was conducted again, and 12 h fasting blood samples and stool specimens were collected at months 0, 3, and 6. Additionally, participants were taught how to complete 3-day food records (FRs) and physical activity logs that were completed one week prior to each study visit. Participants were asked to stop taking any dietary supplements and maintain usual eating habits with the exception of avoiding foods very rich in ACNs and fermented dairy products containing bifidobacteria or lactobacilli. The proposed project and its procedures were reviewed and approved by the University of Connecticut Institutional Review Board (#HR20-0035) prior to the initiation of the project. 

### 2.2. Anthropometric Assessments

A physical examination was conducted at months 0, 3, and 6 to measure height, weight, blood pressure, waist circumference (WC), and body mass index (BMI, kg/m^2^). Body composition data, including lean tissue, body fat, fat mass index (FMI), android fat, gynoid fat, android fat/gynoid fat ratio (A/G ratio), and total fat mass/total lean mass (TFM/TLM), were collected from DXA scans at months 0 and 6.

### 2.3. Assessment of Fasting Blood Lipids and Biomarkers of Inflammation, Oxidative Stress, and Antioxidant Status

#### 2.3.1. Blood Collection and Pretreatment

Fasting blood was collected at months 0, 3, and 6. The blood (80 mL) from 12 h fasting samples was used to determine fasting blood lipid concentrations and biomarkers of inflammation, oxidative stress, and antioxidant status. Plasma samples were collected in EDTA and heparin tubes (BD Vacutainer, Mississauga, ON, Canada). Whole blood samples were centrifuged at 3000× *g* for 15 min at 4 °C, and then, 500 uL of serum and plasma were aliquoted and stored at −80 °C until analyzed. 

#### 2.3.2. Measurement of Fasting Blood Lipids

Changes in total cholesterol (TC), high-density lipoprotein (HDL), and triglycerides (TG) at months 0, 3, and 6 were measured in EDTA plasma using a Cobas c111 analyzer (Roche Diagnostics, Indianapolis, IN, USA). The Friedewald LDL equation was used to determine LDL concentration [19].

#### 2.3.3. Measurement of Blood Biomarkers of Inflammation and Oxidative Stress

Changes in interleukin-1β (IL-1β; BD Biosciences, Franklin Lakes, NJ, USA), thiobarbituric acid reactive substances (TBARS; R&D Systems, Minneapolis, MN, USA), and oxidized low-density lipoprotein (oxLDL; MyBioSource, San Diego, CA, USA) at months 0, 3, and 6 were measured in heparin plasma (for IL-1β) and EDTA plasma (for TBARS and oxLDL) using commercially available ELISA kits. High-sensitivity C-reactive protein (hs-CRP) changes were measured at months 0, 3, and 6 in serum using a Cobas c111 analyzer (Roche Diagnostics, Indianapolis, IN, USA).

#### 2.3.4. Measurement of Blood Biomarkers of Antioxidant Status

Total antioxidant capacity (TAC) in plasma samples was measured using the 2,2′-azino-bis-3-ethylbenzthiazoline-6-sulfonic acid (ABTS) radical chromogen assay [20]. First, a PBS solution was prepared and combined with 2,2′-azobis-(2-amidinopropane) dihydrochloride (AAPH) and ABTS to create the radical solution. After heating, the radical solution was filtered and adjusted using PBS at 734 nm. Samples were mixed with the radical solution, and after 10 min, the absorbance was measured at 734 nm. TAC was calculated using the reduction in absorbance and expressed as mg vitamin C equivalent (VCE)/100 mL. Six-month changes in catalase (CAT) were measured in EDTA plasma using a commercially available ELISA kit (Cayman Chemical, Ann Arbor, MI, USA). 

### 2.4. Statistical Analysis

All significances of variables in baseline characteristics were tested by ANOVA or Chi-square/Fisher’s exact test. All significances of outcome variables were tested using repeated-measures ANOVA with the factors of treatment (0, 392, or 784 mg/day BC) and time (baseline, 3 months, and 6 months) as between-subject and within-subject factors, respectively. Pearson correlations were used to assess the correlation of changes in biomarkers over 6 months. Data are reported as mean ± standard deviation (SD) unless specified otherwise. All data were analyzed using SAS (Version 9.4, SAS Institute, Cary, NC, USA), and *p* < 0.05 was considered statistically significant.

## 3. Results

### 3.1. Baseline

Of the 51 eligible participants for the study, 38 completed the trial and were included in the final analyses (Figure 1). A summary of baseline characteristics of study participants adapted from previously published data [18] is shown in Table 1. Overall, there were no significant differences in age, race, menopause status, and anthropometric measures among the three groups. No differences were observed in dietary or physical activity data among study participants [18]. Additionally, lipid profiles, biomarkers of inflammation, oxidative stress, and antioxidant status were not significantly different at baseline, except for increased TG in the low BC group (*p *< 0.05) and TBARS in the high BC group (*p* < 0.05). The compliance with the provided treatment was ≥95% on average for all three groups [18].

### 3.2. Anthropometric Measurements

BC treatment groups and the control group showed no significant differences in blood pressure and body composition measures at baseline, 3 months, and 6 months. There was a significant time effect for WC (*p* < 0.05); however, there is no significant treatment effect or interaction of time and treatment (Table 2). 

### 3.3. Lipid Profile

A significant treatment effect was observed for TG (*p* < 0.05) (Table 3). Six-month percent changes in TG decreased in a dose-dependent manner with an 11.66% increase in the control group compared to a 5.36% increase and −5.23% decrease in the low BC and high BC groups, respectively. Correlations between six-month changes in biomarkers are shown in Table 4. Six-month changes of TG and TBARS were inversely correlated (R = −0.3328, *p* < 0.05), while the change in LDL was positively correlated with the TC change (R = 0.9191, *p* < 0.0001). 

### 3.4. Inflammatory Biomarkers

A treatment and time interaction was observed for IL-1β (*p* < 0.05) (Table 3). Six-month percent changes in IL-1β displayed a dose-dependent decrease with a 2.82% increase in the control group relative to -1.43% and −11.44% decrease in the low BC and high BC groups, respectively. Six-month changes in IL-1β and hs-CRP were positively correlated (R = 0.4417, *p* < 0.01), and the TG change was also positively correlated with the hs-CRP change (R = 0.4168, *p* < 0.01) (Table 4). 

### 3.5. Biomarkers of Oxidative Stress, Antioxidant Capacity, and Antioxidant Enzymes

A significant treatment effect was observed for TBARS (*p* < 0.05). However, the effects of time and the interaction between treatment and time did not reach statistical significance. Six-month percent changes in TBARS showed a dose-dependent decrease with a 19.2% increase in the control group compared to a 6.09% increase and −2.11% decrease in the low BC and high BC groups, respectively. Mean plasma concentrations of CAT increased overtime in all groups (*p* < 0.05) (Table 3). Six-month changes in both TAC and CAT were inversely correlated with a change in oxLDL (R = −0.3422, *p* < 0.05 and *R* = −0.3392, *p* < 0.05, respectively) (Table 4). 

## 4. Discussion

It has been well documented that CVD risk increases in women during the menopause transition and is related to endogenous sex hormone alterations and adverse changes in their body fat distribution, lipid profile, and measures of vascular health [5]. However, data for the prevention of atherosclerotic CVD and improved survival with lipid-lowering interventions are lacking and remain elusive for women with an elevated risk. Non-pharmaceutical clinical trials are beneficial for determining both safe and effective dietary strategies that may reduce the risk of CVD while having minimal side effects, as compared to drugs, and thus greater adherence. The current study suggests that BC ACN may mitigate the risk of CVD in peri- and early postmenopausal women through improving blood lipids, inflammation, and lipid peroxidation as indicated by decreasing trends in blood TG, IL-1β, and TBARS levels in a dose-dependent manner, and significant inverse correlations between six-month changes in CAT and TAC with oxLDL. No significant changes to body composition were observed over the six-month intervention period, except for WC. Although WC showed differences on a timewise basis, there were no significant alterations in all other measures of body composition. Body fat composition and distribution play an important role as risk factors for CVD. A recent study suggests that breast fat accumulation may be predictive of major adverse cardiovascular events (MACE) in premenopausal women [21]. Breast fat has also been shown to act as a potential new independent risk factor for CVD and worsened cardiac performance mediated by over-inflammation and activation of SGLT-2/SIRT-3 pathways [22]. Further analysis of body composition measures in a study population with a broader range of age and menopause status is needed to determine the effects of BC on measures of body composition, including breast fat accumulation.

TG are a well-established biomarker of CVD risk, as evidenced by epidemiological studies [23,24,25], genome-wide analyses [26,27,28], and Mendelian randomization studies [29,30]. Six-month BC supplementation resulted in a significant decrease in TG levels in a dose-dependent manner. This result is consistent with an intervention using grape powder (10 g/day) in postmenopausal women that found lowered serum TG levels following supplementation for 21 days [31]. In a meta-analysis on the effectiveness of ACN modifications of cardiometabolic risk factors, ACNs were found, in some cases, to contribute to reductions in TG, TC, and LDL in healthy adults, but they were most effective in adults with dyslipidemia [32]. Although the TG-lowering mechanisms of ACN are not fully understood, it has been attributed to reductions in apolipoprotein B and C111 in diabetic patients [33]. Researchers have suggested that the ability of ACN to reduce TC can potentially be related to enhanced excretion of bile acids and decreased sterol retention [34], while LDL lowering abilities may be via inhibition of cholesteryl ester transfer protein and upregulation of the LDL-receptor [35]. BC ACN appear to have contributed to the decreasing trend of TG shown in the current study; however, further research is needed to elucidate these mechanisms and determine the magnitude of the effect of BC ACN supplementation in women with elevated lipid measures.

Inflammation plays a critical role in the initiation and progression of CVD [36]. In particular, pro-inflammatory cytokines play a key role in atherosclerotic plaque formation [37]. The present study found a reducing effect of BC on IL-1β, which is dependent on both treatment group and time, and six-month percent changes in IL-1β showed a dose-dependent decreasing trend between groups. Thus, this suggests that the anti-inflammatory potential of BC through the suppression of the pro-inflammatory cytokine, IL-1β. In our previous in vitro study, BC repressed the expression of IL-1β in RAW 264.7 macrophages treated with lipopolysaccharide [17]. Furthermore, cyanidin and delphinidin, two major BC ACN, have been shown to have anti-inflammatory properties, and one mechanism is thought to be through the inhibition of translocation of the nuclear factor κB (NF-κB), which, consequently, can prevent the gene expression of pro-inflammatory cytokines [38]. Additionally, our in vivo studies with animal models showed that in humerus homogenates of aged mice, BC increased CAT activity and significantly reduced elevated concentrations of TNFα, a pro-inflammatory cytokine that contributes to CVD risk [39]. 

A significant reduction in TBARS, a biomarker of lipid peroxidation, was seen among treatment groups in a dose-dependent manner. A study evaluating the effectiveness of BC and chokeberry for preventing lipid membrane peroxidation found that BC and chokeberry were effective in preventing lipid peroxidation [40]. The authors suggested that the protective effects are presumably due to the antioxidant activity of the extracts owing to cyanidin derivatives, which are effective in protecting the lipid membrane. The current study also found an inverse correlation between TG and TBARS. Previously, an analysis among pre- and postmenopausal women in the general population found a positive association between TG and TBARS [41]. Authors explained that the primary substrate for lipid peroxidation is polyunsaturated fatty acids, and TG may act as a substrate and a determinant of the amount of TBARS produced [42], which may explain this result. The differing results in the current study may be due to the significant differences between groups in TG and TBARS at baseline, suggesting that further research in a larger, more diverse sample is needed to fully understand the relationship between these two measures.

Decreased production of estrogen, a natural antioxidant, during menopause can create a pro-oxidant state [43,44], which is a common etiology of CVD. The major BC ACN include cyanidin-3-glucoside (C3G), cyanidin-3-rutinoside (C3R), delphinidin-3-glucoside (D3G), and delphinidin-3-rutinoside (D3R) [9]. Cyanidin and delphinidin have been reported to have phytoestrogen activity [45], which may be the driving factor behind BCs antioxidative capacity in this study population. Additionally, a recent in vitro study found that food-derived C3G aided in restoring the antioxidant enzyme activities of SOD and CAT [46]. Antioxidant mechanisms are imperative for scavenging reactive oxygen species (ROS) and reducing the oxidation of cellular molecules. In the current study, changes in TAC, a measure of radical scavenging activity, and CAT were inversely associated with changes in oxLDL, which altogether suggests the potential of BC as an effective dietary strategy for preventing lipid oxidation through maintaining the antioxidant defense mechanisms.

While CAT was significantly increased in the BC treatment groups, it was also significantly increased in the control group. This could be due to the influence of vitamin D on antioxidant enzymes, including CAT. Although the antioxidant potential of vitamin D is not fully understood [47], vitamin D may have indirect antioxidant effects through influencing the expression of genes and proteins involved in oxidative stress. Particularly, it has been shown that the vitamin D receptor plays an important role in protecting cells from excessive production of ROS and, thus, preventing cell damage [48]. It has been further observed by Piotrowska et al. that vitamin D and its analogs regulate the production of ROS and expression of ROS-associated genes such as CAT [49]. This suggests that the increase observed in CAT across all groups may be due to the influence of vitamin D.

The current study has several strengths, including that this study targeted women during menopause transition, which is a critical period to begin intervention for the prevention of CVD, and the compliance with the provided treatment was, on average, ≥95% for all groups. Some limitations include that only healthy peri- and early postmenopausal women were included in the study, and the findings are not applicable to general adult female populations. Second, the study was designed to investigate the effects of BC on the reduction of osteoporosis risk, and the initial screening did not include the collection of reproductive history, including pregnancies and any adverse cardiovascular events, such as pre-eclampsia and hypertensive disorders, which can place women at long-term risk for CVD development [50]. Third, there is a potential influence of vitamin D and calcium supplementation on the measured biomarkers. Vitamin D has been shown to possess anti-inflammatory and antioxidative properties [51]; however, it has also been shown to have no confirmed additional benefits for cardiovascular health [52], and the effect of calcium supplementation on the lipid profile remains inconsistent [53]. Lastly, the lack of diversity in the study population reduces generalizability, and the small sample size of this pilot study might reduce the power to detect significant changes and dose-dependent responses in some of the biomarkers investigated.

## 5. Conclusions

Overall, the findings of this study support our hypothesis that daily BC consumption for six months potentially mitigates CVD risk in peri- and early postmenopausal women through improving dyslipidemia, inflammation, and lipid peroxidation, as shown by decreasing trends of TG, IL-1β, and TBARS following BC treatment. Significant inverse associations observed between CAT and TAC changes with oxLDL changes further suggest that BC supplementation may aid in mitigating CVD through its antioxidant properties. Additional microbial and metabolomics analyses may elucidate mechanisms of action for BC on the modification of lipids and the reduction of oxidative stress and inflammation through the gut and its microbial metabolites such as short-chain fatty acids, bile acids, sterols, enterolactone, protocatechuic acid, and gallic acid [54]. An investigation of additional biomarkers may reveal protective mechanisms of BC not only for reducing the risk of CVD but also for amelioration of other chronic disease risk factors and hepatic function. Further longer-term investigations, especially with larger samples and adult women of a broader age range at a higher risk for CVD, are warranted to confirm these promising findings.

## Figures and Tables

**Figure 1 biomedicines-11-02834-f001:**
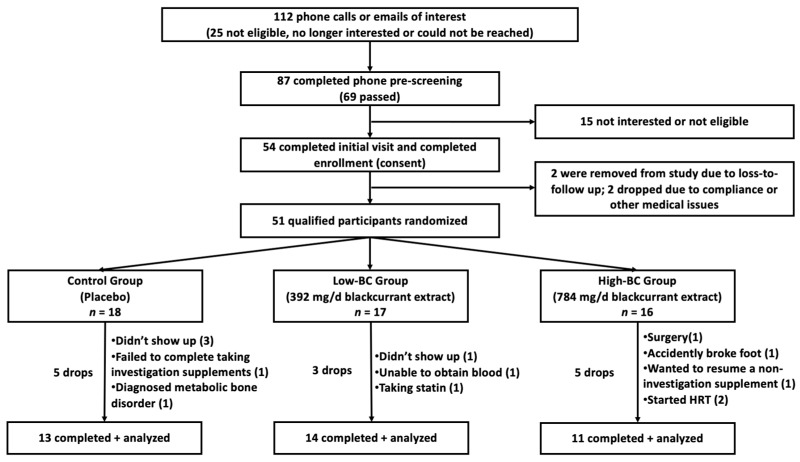
Study design, inclusion, and follow-up of study participants. Final analysis *n* = 38. Adapted from previously published data [18].

**Table 1 biomedicines-11-02834-t001:** Participant characteristics at baseline.

	All (*n* = 38)	Control (Placebo, *n* = 13)	Low BC (392 mg/day, *n* = 14)	High BC (784 mg/day, *n* = 11)	*p*-Value
Age, (y)	52.9 ± 4.2	54.3 ± 3.8	53.1 ± 4.6	50.9 ± 4.2	0.155
Race/ethnicity, *n* (%)					
Caucasian	35 (92.1)	13 (100)	11 (78.6)	11 (100)	0.272
Hispanic	1 (2.6)		1 (7.1)		
Asian American	2 (5.3)		2 (14.3)		
Menopause, *n* (%)	26 (68.4)	8 (61.5)	11 (78.6)	7 (63.6)	0.677
SBP (mmHg)	114.5 ± 14.6	115.7 ± 14.1	117.0 ± 15.9	109.9 ± 13.5	0.458
DPB (mmHg)	78.9 ± 7.6	80.5 ± 7.1	79.4 ± 8.3	76.5 ± 7.4	0.432
WC (cm)	84.3 ± 13.5	85.5 ± 15.6	85.2 ± 12.4	81.8 ± 12.0	0.758
BMI (kg/m^2^)	26.5 ± 5.7	27.0 ± 7.2	26.4 ± 4.5	25.8 ± 4.9	0.878
Lean tissue %	60.1 ± 6.6	61.7 ± 7.2	59.3 ± 6.3	61.0 ± 6.1	0.616
Body fat %	36.1 ± 7.0	35.0 ± 7.7	37.5 ± 6.6	35.5 ± 6.5	0.608
FMI (kg/m^2^)	9.8 ± 4.0	9.9 ± 4.9	10.1 ± 3.2	9.4 ± 3.5	0.909
Android fat %	37.2 ± 11.2	34.9 ± 12.3	40.3 ± 11.3	35.9 ± 9.0	0.412
Gynoid fat %	42.1 ± 5.8	41.1 ± 5.8	42.8 ± 5.1	42.3 ± 6.4	0.728
A/G ratio	0.9 ± 0.2	0.8 ± 0.2	0.9 ± 0.2	0.8 ± 0.1	0.280
TFM/TLM	0.6 ± 0.2	0.6 ± 0.2	0.7 ± 0.2	0.6 ± 0.2	0.666
TC (mg/dL)	183.3 ± 35.7	178.1 ± 31.0	179.7 ± 34.4	194.0 ± 42.0	0.501
HDL (mg/dL)	67.8 ± 15.6	71.5 ± 17.2	62.0 ± 15.3	70.8 ± 13.9	0.227
LDL (mg/dL)	97.96 ± 31.0	92.86 ± 25.70	95.16 ± 28.95	107.56 ± 38.49	0.475
TG (mg/dL)	87.6 ± 40.5	68.4 ± 30.2	112.8 ± 57.0	78.1 ± 20.7	<0.05
hs-CRP (mg/L)	2.0 ± 2.6	1.3 ± 1.6	2.8 ± 3.7	1.6 ± 1.6	0.304
IL-1β (pg/mL) *	17.0 ± 6.9	18.0 ± 5.8	17.5 ± 7.6	15.3 ± 7.4	0.679
TBARS (uM)	0.61 ± 0.2	0.51 ± 0.2	0.59 ± 0.2	0.76 ± 0.2	<0.05
oxLDL (ng/mL)	236.0 ± 59.1	223.3 ± 58.6	242.5 ± 56.9	242.8 ± 62.2	0.635
TAC (mg VCE/100 mL)	9.62 ± 1.63	9.99 ± 1.59	9.38 ± 1.09	9.50 ± 2.18	0.606
CAT (nmol/min/mL)	3.37 ± 1.41	3.53 ± 1.65	3.00 ± 1.05	3.66 ± 1.52	0.461

Values displayed as mean ± SD or *n* (%). * Analysis was performed on 35 participants for IL-1β due to missing values. SBP, systolic blood pressure; DPB, diastolic blood pressure; WC, waist circumference; BMI, body mass index; FMI, fat mass index; A/G ratio, android/gynoid fat ratio; TFM/TLM, total fat mass/total lean mass; TC, total cholesterol; HDL, high-density lipoprotein; LDL, low-density lipoprotein; TG, triglycerides; hs-CRP, high sensitivity C-reactive protein; IL-1β, interleukin-1beta; TBARS, thiobarbituric acid reactive substances; oxLDL, oxidized low-density lipoprotein; TAC; total antioxidant capacity; VCE, vitamin C equivalent; and CAT, catalase.

**Table 2 biomedicines-11-02834-t002:** Anthropometric characteristics of study participants at baseline, three months, and six months.

Marker	Month				*p*-Value		
		Control (Placebo, *n* = 13)	Low BC (392 mg/Day, *n* = 14)	High BC (784 mg/Day, *n* = 11)	Group (Low Dose vs. High Dose vs. Placebo	Time (Baseline vs. 3 Months vs. 6 Months)	Time × Group (Interaction)
SBP (mm Hg)	Baseline	115.7 ± 4.1	117.0± 3.9	109.9.8 ± 4.4	0.798	0.931	0.279
	3	115.6 ± 4.2	111.8 ± 4.0	113.3 ± 4.6			
	6	115.2 ± 3.6	114.0 ± 3.4	112.6 ± 3.8			
DBP (mmHg)	Baseline	80.5 ± 2.1	79.4± 2.0	76.5 ± 2.3	0.632	0.918	0.777
	3	79.9 ± 2.6	77.8 ± 2.5	77.6 ± 2.8			
	6	79.6 ± 2.8	80.4 ± 2.7	76.8 ± 3.0			
BMI (kg/m^2^)	Baseline	27.0 ± 1.6	26.4 ± 1.5	25.8 ± 1.7	0.848	0.100	0.787
	3	27.4 ± 1.6	26.6 ± 1.6	26.1 ± 1.8			
	6	27.3 ± 1.6	26.5 ± 1.6	25.8 ± 1.8			
WC (cm)	Baseline	86.5 ± 3.7	85.2 ± 3.6	81.8 ± 4.1	0.738	<0.01	0.578
	3	86.1 ± 3.8	87.5 ± 3.7	83.4 ± 4.2			
	6	86.3 ± 3.8	88.2 ± 3.6	83.5 ± 4.1			
Lean tissue %	Baseline	61.7 ± 1.8	59.3 ± 1.7	61.0 ± 2.0	0.720	0.681	0.609
	6	61.1 ± 2.0	59.5 ± 1.9	60.9 ± 2.1			
Body fat %	Baseline	35.0 ± 1.9	37.5 ± 1.9	35.5 ± 2.1	0.711	0.689	0.564
	6	35.6 ± 2.1	37.3 ± 2.0	35.6 ± 2.3			
FMI (kg/m^2^)	Baseline	9.9 ± 1.1	10.1 ± 1.1	9.4 ± 1.2	0.910	0.476	0.551
	6	10.1 ± 1.2	10.1 ± 1.1	9.4 ± 1.3			
Andriod fat %	Baseline	34.9 ± 3.1	40.3 ± 3.0	35.9 ± 3.3	0.511	0.884	0.768
	6	35.3 ± 3.4	39.7 ± 3.3	36.3 ± 3.7			
Gynoid fat %	Baseline	41.1 ± 1.6	42.8 ± 1.5	42.3 ± 1.7	0.867	0.774	0.387
	6	42.0 ± 1.7	42.7 ± 1.7	41.9 ± 1.9			
A/G ratio	Baseline	0.8 ± 0.0	0.9 ± 0.0	0.8 ± 0.1	0.358	0.860	0.763
	6	0.8 ± 0.1	0.9 ± 0.1	0.8 ± 0.1			
TFM/TLM	Baseline	0.6 ± 0.1	0.7 ± 0.1	0.6 ± 0.1	0.771	0.436	0.631
	6	0.6 ± 0.1	0.6 ± 0.1	0.6 ± 0.1			

Data are mean ± SD. SBP, systolic blood pressure; DPB, diastolic blood pressure; WC, waist circumference; BMI, body mass index; FMI, fat mass index; A/G ratio, android fat/gynoid fat ratio; and TFM/TLM, total fat mass/total lean mass.

**Table 3 biomedicines-11-02834-t003:** Fasting blood lipids and biomarkers of inflammation, oxidative stress, and antioxidant status in study participants at baseline, three months, and six months (*n* = 38).

Marker	Month				*p*-Value		
		Control (Placebo, *n* = 13)	Low BC (392 mg/Day, *n* = 14)	High BC (784 mg/Day, *n* = 11)	Group (Low Dose vs. High Dose vs. Placebo	Time (Baseline vs. 3 Months vs. 6 Months)	Time × Group (Interaction)
TC (mg/dL)	Baseline	178.1 ± 9.9	179.7 ± 9.5	194.0 ± 10.8	0.330	0.658	0.205
	3	180.5 ± 9.4	177.5 ± 9.1	196.7 ± 10.2			
	6	184.0 ± 10.1	167.6 ± 9.7	194.2 ± 11.0			
HDL (mg/dL)	Baseline	71.5 ± 4.3	62.0 ± 4.2	70.8 ± 4.7	0.161	0.807	0.516
	3	72.3 ± 4.8	62.9 ± 4.7	71.6 ± 5.3			
	6	72.8 ± 4.6	59.9 ± 4.5	73.5 ± 5.0			
LDL (mg/dL)	Baseline	92.9 ± 4.8	95.2 ± 4.7	107.6 ± 5.2	0.315	0.734	0.406
	3	93.1 ± 4.2	93.3 ± 4.2	107.6 ± 4.7			
	6	96.3 ± 4.5	85.3 ± 4.3	107.0 ± 4.9			
TG (mg/dL)	Baseline	68.4 ± 11.2	112.8 ± 10.8	78.1 ± 12.2	<0.05	0.507	0.134
	3	75.5 ± 10.0	106.5 ± 9.6	87.4 ± 10.9			
	6	74.2 ± 9.2	112.3 ± 8.8	68.9 ± 10.0			
hs-CRP (mg/L)	Baseline	1.3 ± 0.7	2.8 ± 0.7	1.6 ± 0.8	0.271	0.622	0.355
	3	1.2 ± 0.8	3.1 ± 0.8	1.3 ± 0.9			
	6	1.3 ± 0.6	2.2 ± 0.6	1.6 ± 0.6			
IL-1β (pg/mL) *	Baseline	18.0 ± 2.0	17.5 ± 2.8	15.4 ± 2.3	0.636	0.663	<0.05
	3	17.9 ± 2.0	17.4 ± 1.8	15.4 ± 2.3			
	6	18.4 ± 2.0	17.3 ± 2.8	15.2 ± 2.3			
TBARS (uM)	Baseline	0.50 ± 0.04	0.59 ± 0.03	0.76 ± 0.04	<0.05	0.997	0.140
	3	0.65 ± 0.03	0.54 ± 0.03	0.65 ± 0.04			
	6	0.53 ± 0.04	0.56 ± 0.03	0.73 ± 0.04			
oxLDL (ng/mL)	Baseline	216.9 ± 9.4	242.5 ± 8.7	242.9 ± 9.9	0.861	0.694	0.489
	3	228.6 ± 8.7	252.7 ± 8.7	244.3 ± 9.9			
	6	238.7 ± 7.9	233.8 ± 7.4	239.4 ± 8.3			
TAC(mg VCE/100 mL)	Baseline	9.9 ± 0.3	9.4 ± 0.2	9.5 ± 0.3	0.282	0.657	0.420
	3	10.0 ± 0.2	9.6 ± 0.2	8.7 ± 0.3			
	6	10.3 ± 0.3	9.3 ± 0.2	9.6 ± 0.3			
CAT (nmol/min/mL)	Baseline	3.5 ± 0.2	3.0 ± 0.2	3.7 ± 0.2	0.141	<0.01	0.384
	3	3.4 ± 0.3	3.8 ± 0.3	5.0 ± 0.3			
	6	4.0 ± 0.3	4.5 ± 0.3	5.3 ± 0.3			

* Analysis was performed on 35 participants for IL-1β due to missing values. TC, total cholesterol; HDL, high-density lipoprotein; LDL, low-density lipoprotein; TG, triglycerides; hs-CRP, high sensitivity C-reactive protein; IL-1β, interleukin-1beta; TBARS, thiobarbituric acid reactive substances; oxLDL, oxidized low-density lipoprotein; TAC, total antioxidant capacity; VCE, vitamin C equivalent; and CAT, catalase.

**Table 4 biomedicines-11-02834-t004:** Pearson correlation coefficients for 6-month changes in blood lipids and biomarkers of inflammation, oxidative stress, and antioxidant status.

	∆TC	∆HDL	∆LDL	∆TG	∆hs-CRP	∆IL-1β	∆TBARS	∆oxLDL	∆TAC	∆CAT
∆TC	1.0									
∆HDL	0.57061 (0.0002)	1.0								
∆LDL	0.91909 (<0.0001)	0.28388 (0.0841)	1.0							
∆TG	−0.02880 (0.8637)	−0.13880 (0.4059)	−0.24332 (0.1410)	1.0						
∆hs-CRP	0.25663 (0.1199)	0.24334 (0.1410)	0.08352 (0.6181)	0.41675 (0.0092)	1.0					
∆IL-1β	0.10998 (0.5110)	0.03787 (0.8214)	0.04476 (0.7896)	0.24585 (0.1368)	0.44174 (0.0055)	1.0				
∆TBARS	−0.06238 (0.7099)	−0.02323 (0.8899)	0.02688 (0.8728)	−0.33275 (0.0412)	−0.11608 (0.4877)	0.00680 (0.9677)	1.0			
∆oxLDL	0.02847 (0.8653)	−0.09624 (0.5654)	0.05442 (0.7458)	0.06411 (0.7022)	0.20878 (0.2084)	−0.08622 (0.6068)	−0.16624 (0.3185)	1.0		
∆TAC	−0.06189 (0.7121)	−0.04986 (0.7663)	−0.11270 (0.5005)	0.23321 (0.1588)	0.15710 (0.3462)	0.08225 (0.6235)	−0.07633 (0.6488)	−0.34222 (0.0355)	1.0	
∆CAT	−0.07445 (0.6569)	0.13326 (0.4216)	−0.17861 (0.2833)	0.14689 (0.3788)	−0.05463 (0.7446)	−0.05902 (0.7249)	−0.09861 (0.5599)	−0.33915 (0.0372)	0.11525 (0.4908)	1.0

Data are Pearson correlation coefficient (*p*-value). TC, total cholesterol; HDL, high-density lipoprotein; LDL, low-density lipoprotein; TG, triglycerides; hs-CRP, high sensitivity C-reactive protein; IL-1β, interleukin-1beta; TBARS, thiobarbituric acid reactive substances; oxLDL, oxidized low-density lipoprotein; TAC, total antioxidant capacity; and CAT, catalase.

## Data Availability

Data is contained within the article.

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
