# Peer review of "Blackcurrant Anthocyanins Improve Blood Lipids and Biomarkers of Inflammation and Oxidative Stress in Healthy Women in Menopause Transition without Changing Body Composition"

_biomedicines, 2023, doi:10.3390/biomedicines11102834_

Round 1
Reviewer 1 Report
This was a study of blackcurrent anthocyanins in women about the stage of menopause.
There is some earlier data to support the rationale for the study.
There are several issues with the study:
Sample size – the final sample size was about 10 women in each group – there are many issues with such a sample size most importantly that any changes can arise by chance – this is indicated by the observation that 2 parameters were statistically significantly different at baseline i.e. randomisation did not produce equivalent starting groups. There should be some power calculations and also the issues of sample size should be front and centre in the Discussion.
Dose – it is excellent science that the authors have chosen more then one dose of the test material – dose dependency is an essential component of science - however in pharmacology doses should be based on log not arithmetic rages – the low and high dose varied by 2 fold – it would be been better if they varied by half or one log unit. There should be more discussion of the dose dependency in the results.
Much of the results and discussion id about statical significance – but the first element of scientific data is the size of the change – so date should be discussed as the size of the changes (absolute, %, fold etc) and then comment on the statistical significance.
The study looked at biomarkers of CBD and nor CVD events ( difficult in a 6 moth study) – so the stamen “Second, to our knowledge this is the first human clinical trial to investigate 334 the effects of BC supplementation for the prevention of CVD in this high risk population- 335 tion” – studies are generally not publishable if they are not the first so this can be omitted and the study did not look at the PREVENTION of CVD only changes in biomarkers.
The paper also uses anti-inflammatory and anto0oxidant very uncritically and non-specifically – inflammation and oxidation are complex responses with many many individual elements – the authors should be more specific in mention actual specific responses not the catch all of inflammation and oxidation.
Results – there is a high amount of negative or “no change” data – perhaps this could be moved to a supplement and just the relevant data with changes highlighted in a table.
Did the paper provide information on concurrent therapies particularly statins and blood pressure medication.
END
Reviewer 2 Report
This manuscript is an ambitious work with many healthy women. It is important to develop new compound as supplement for extension of healthy life expectancy. Blackcurrant anthocyanin is a potential candidate. However, the effect of BC ACN supplementation did not show any significant differences except decreasing in TG concentrations and increasing in catalase activities in this study.
I have any questions as follows.
1. Did you measure plasma insulin and adiponectin concentrations?
2. Did you measure plasma total albumin concentrations and some hepatic deviation enzymes (AST, ALT, LDH and others) activities?
3. Did you measure plasma estrogen concentrations?
4. How do you think about the effect of higher dose and longer duration with BC CAN supplementation?
Anthocyanin-rich compounds show significant anti-oxidative and anti-inflammatory capacity in a dose-response manner in subjects with dyslipidemia (Redox Biol. 2020, 32, 101474; Am J Clin Nutr. 2019, 109, 1535-45). In this study, subjects were healthy and non-obese women. So, it is considered that the effect of BC CAN supplementation was no apparent.
Effects with BC CAN supplementation are considered to amelioration of hepatic function and improvement of lipid metabolism, and the effects are not specific to reducing the risk of cardiovascular diseases. Authors should rewrite the discussion.
Round 2
Reviewer 2 Report
This manuscript is revised well according to the reviewers' comments.
This manuscript is improved to be accepted for publication.
Author Response
Thank you very much for your favorable comments. With your review we were able to improve the clarity and quality of the manuscript. The manuscript has been further revised by adding comments from the academic editor.